# Dribble Accuracy and Arm Coordination Pattern According to Motor Expertise and Tempo

**DOI:** 10.3390/ijerph20105788

**Published:** 2023-05-11

**Authors:** Jinhan Park, Jaeuk Jeong

**Affiliations:** 1Department of Applied Physiology and Kinesiology, University of Florida, Gainesville, FL 32611, USA; 2Department of Physical Education, College of Education, Seoul National University, Seoul 08826, Republic of Korea

**Keywords:** motor control, dribble accuracy, coordination pattern, motor expertise, tempo

## Abstract

Skilled movements in motor learning result from efficiently controlling the many degrees of freedom in human movement. To acquire motor skills, harmonious coordination of body segments in time and space is crucial for accurate and consistent performance. The purpose of this study was to compare dribbling accuracy, consistency, and coordination patterns of body segments according to motor expertise and tempo. To achieve this, we had eight basketball experts and eight beginners perform static dribbling at three different speeds for 20 s. Force plates measured radial error while motion capture equipment measured the angular data of the right arm’s fingers, wrist, and elbow. The measurements obtained from the force plate were used to analyze the participants’ dribbling performance, including accuracy, consistency, and coordination patterns. The research results showed that there was no significant difference in dribbling accuracy according to skill level, but skilled players showed higher consistency in the anterior–posterior (AP) direction (*p* < 0.001). In the comparative analysis of coordination patterns, skilled players showed an in-phase structure, whereas beginners showed an anti-phase structure (elbow–wrist: *p* < 0.05; wrist–finger: *p* < 0.001; elbow–finger: *p* < 0.001). This study suggests that achieving proficiency in basketball dribbling requires a strategy that involves coordination of movements with an in-phase pattern for stability in performance.

## 1. Introduction

In basketball, dribbling is a fundamental skill that involves rhythmically bouncing the ball. It is crucial for a basketball player to master this motor skill in order to move around freely on the court. However, in competitive sports situations, dribbling can be challenging due to various internal and external factors, such as arm coordination and the positions of teammates and defenders. To avoid defenders, a skilled player must have proper arm coordination and be able to dribble the ball with different tempos.

Studies on basketball dribbling have highlighted the importance of players having enough energy to bounce the ball on the ground and absorb the power from its rebound. Mohamed and his colleagues developed a human dribbling model based on the performance of professional basketball players, which suggests that skilled arm coordination involves properties of dumping and duffing stiffness to control the bouncing energy [1]. Compared to novice basketball players, this type of arm coordination leads to longer contact time between the hand and ball, which effectively minimizes the energy and contributes to higher dribbling accuracy and consistency [2].

Recent studies [3,4] have found that the dribbling arm coordination pattern differs depending on the level of expertise. Professional basketball players showed lower variability in their wrist and higher variability in their elbow and shoulder compared to amateur players. According to these studies, the increased variability in the elbow and shoulder of professional basketball players might be an adaptive strategy for coping with changing environmental factors during a basketball game. This variability allows them to adjust their arm movements and dribbling techniques to adapt to different situations, such as changes in the position of defenders or teammates and different court locations.

Katsuhara and his colleagues [2] suggest that the differences in dribbling performance between experts and novices can provide valuable kinematic information for novice learners to improve their dribbling skills. However, they suggest that simply analyzing isolated joint movements and their variability is not enough to fully explain the complexities of dribbling. Therefore, it is necessary to examine the combinations of multiple degrees of freedom (DOF) that are created by the arm joints working together. In the motor control area, researchers need to study how the large number of DOF are organized in human movement [5,6,7,8]. The organization of DOF refers to how the many joints and muscles in the human body work together to produce coordinated movements. By studying this organization, researchers can develop theoretical models of motor control and design experiments to test these models. For example, in a study examining the movement of the upper limb joints in tennis serving from the perspective of DOF and coordination, it was reported that the connectivity of the kinetic chain initiated from the proximal joint is related to the stability of the movement [9]. This approach can provide insights into the complex processes underlying human movement and inform the development of interventions to improve motor performance. However, research on the cooperative movements that occur in various combinations of arm joints during dribbling performance has not been adequately conducted.

According to the dynamic system approach, the human motor system has the ability to maintain a stable state despite changes in the environment. However, altering environmental factors, such as dribbling tempo, can break the stability of the human motor system. This means that the pattern of arm coordination in dribbling may change depending on the tempo of the task. Kelso and his colleagues suggest that increasing task speed over a critical point can lead to the emergence of a new coordination structure, which enhances stability and improves motor performance [10]. In this case, the performer can automatically transition from the original motor system to a new one through self-organization or phase transition. For example, when gradually increasing speed, human locomotion can change from walking to running [11]. In sports, an increase in swimming stroke rate can also alter bimanual coordination, with different patterns of phase transition being shown between experts and novices [12,13].

Studying the harmonious movements of the various joints involved in dribbling and how they change with variations in tempo is essential to gain a deeper understanding of human motor control. This knowledge can help in developing effective training methods and preventing injuries in athletes. However, there is a lack of research on the organic motion of limbs involved in dribbling and their coordination, and existing studies investigating arm coordination during dribbling have limitations in terms of sample size and age range [14]. The limitations of such previous research have a clear constraint in terms of ecological validity, which is insufficient to generalize research results or apply them to the real world. Therefore, to overcome the limitations of prior research and deepen the understanding of human movement, this study aimed to compare the accuracy and consistency of skilled and novice players’ performance in dribbling tasks across different speed conditions and to determine if the differences in kinematic coordination contribute to the underlying mechanism.

Here, we investigated whether the motions of the arm limb joint angles are organizationally modulated in order to achieve accuracy and consistency in dribbling. According to previous research on basketball dribbling tasks, the joint angles of the fingers, wrist, and elbow, excluding the shoulder, showed the highest correlation, while the shoulder joint showed the lowest correlation with skill level. Therefore, in this study, the variables were limited to the coordination structures of the joints of the fingers, wrist, and elbow. Moreover, to analyze the coordination of these joints, the cophase variable was utilized, which was based on fast Fourier transform (FFT) analysis of the angular data for each joint movement. This allowed for the identification of the dominant peak frequency for each joint, which is also known as the modal frequency.

We investigated dribble accuracy and different arm coordination patterns according to different levels of expertise and tempo to understand the internal dynamics of human motor control. We hope to complement the limitations of previous studies and provide useful empirical evidence for basketball learners and coaches in the future. To achieve this, we formulated the following two hypotheses: First, the accuracy and consistency of dribbling will vary according to the direction of movement depending on skill level and tempo. Second, differences and changes in arm coordination patterns will occur depending on skill level and tempo.

## 2. Methods

### 2.1. Participants

We recruited 16 participants based on basketball experience, meeting the statistical power of 95% calculated using the G*Power program for a repeated measures design. Eight were undergraduate students who had no experience in formal basketball training and were grouped into the novice group. The other eight were collegiate basketball athletes who had competed at the Korean national amateur competition and were grouped into the expert group (average experience of 10 years). Participants in both groups showed right-handedness and no significant differences in arm length (novice: 75.9 ± 4.6 cm; expert: 74 ± 3.0 cm, *p* = 0.355). We excluded one participant who had difficulty dribbling a basketball due to his physical and/or neurological condition. In addition, participants who had prior experience in similar experiments were also excluded. All participants were required to provide informed consent, and all research procedures were conducted in compliance with the regulations of the review board.

### 2.2. Materials and Tasks

To measure dribbling accuracy and consistency for the target, force plates (OR6-7-OP, Advanced Mechanical Technology Inc., Waltham, MA, USA) were used. Around the force plate, eight motion capture cameras (Miqus, Qualysis, Göteborg, Sweden) were set up to monitor arm coordination during dribbling. We attached seven reflective markers (diameter: 15 mm) on the participants’ right upper limbs. To control the tempo of dribbling, a metronome was used (Metronome, Panoramic Software Inc., Surrey, BC, Canada).

The task was the high static dribble, which bounces the ball up to waist height without moving. Participants were instructed to dribble the ball as accurately as possible inside a square target (2 × 2 cm) on the force plate according to the metronome tempo. Three tempo conditions (normal, medium, and fast) were defined based on the preferred dribble tempo and the fastest tempo. The normal tempo (NT) was the preferred speed for 20 s of dribbling, and the medium tempo (MT) was defined as 40% between the preferred and the fastest tempo. The fast tempo (FT) was defined as 80% (Figure 1).

During the task, the left foot was maintained 45 degrees forward of the right foot, and the gaze was fixed on a target 2 m ahead. In order to investigate differences in arm coordination patterns, all subjects were asked to restrict movement of their trunk and lower body in order to be as still as possible.

### 2.3. Procedure

When participants arrived at the laboratory, they were adequately briefed about the experiment and performed a sufficient warm-up before the task. To calculate the tempo conditions, each participant’s preferred and fastest dribble tempos were measured in advance. The reference tempo was calculated based on the number of times the ball bounced on the ground for 10 s (5–15 s out of 20 s). After setting the three conditions of tempo, markers were placed on the participant’s right arm (metacarpus, proximal phalanx, middle phalanx, tip of the metacarpus, lateral aspect of the elbow, lateral aspect of the wrist, and acromion) based on previous research [15,16,17]. After attaching the markers and calibrating the 3D motion capture system, participants were instructed to stand next to the force plate and perform the dribble task.

Participants practiced the task for 10 s and then performed 3 trials for each experimental condition. Each trial lasted for 40 s of the dribble task according to the metronome tempo. Participants were given a 5 min rest after each trial to prevent fatigue. Participants performed the task and were instructed as follows: “Maintain a forward gaze and dribble the ball accurately towards the target on the ground while trying to synchronize with the metronome tempo”. The entire session took approximately 1 h. Cases in which the ball was out of the force plate or when there were excessive torso and lower body motions were excluded from the data analysis.

### 2.4. Data Analysis

Data on dribbling accuracy, consistency, and coordination patterns were preprocessed and analyzed using customized codes in MATLAB (MATLAB, Mathworks Inc., Natick, MA, USA). We calculated dribble accuracy and consistency based on the mean radial error (MRE) and the standard deviation (SD) of the MRE, respectively. Radial error (RE) was calculated from the bounce position to the target on the force plates (Formula (1)), and MRE was calculated as the average value during the task (Formula (2)).
(1)$$\mathrm{RE}=\sqrt{x^{2}+y^{2}}$$
(2)$$\mathrm{MRE}=\frac{{\Sigma \mathrm{RE}}_{i}}{n}$$
where *x* is the value of the center of pressure (COP*_x_*) in the anterior–posterior (AP) direction based on body alignment, and *y* is the value of COP*_y_* in the medial–lateral (ML) direction (Figure 2). *n* is the number of bounces for 40 s, and RE*_i_* is the radial error of the *i*th sample. The sampling rate was 100 Hz, and only trials with a ground reaction force of at least 300 N were extracted.

Kinematic data were collected using a motion capture system. We calculated the subjects’ arm angular data of the middle finger, wrist, and elbow joints using a Qualisys Track Manager (QTM, Qualisys, Göteborg, Sweden). We determined the phase angles (*θ*) in the plane of angular velocity and displacement of each dribble. Angular movements were investigated by applying the cophase technique [18]. The cophase was calculated to analyze the coordinate pattern between the arm joints (finger–wrist, wrist–elbow, finger–elbow) as Formula (3), where P*_xy_* represents the cross-power spectral density of two time series among joint movements [19].
(3)$$\mathrm{Cophase}=\mathrm{arc}\tan\left(\frac{-imag\left(p_{xy}\right)}{real\left(p_{xy}\right)}\right)\times \left(\frac{180{}^{\circ}}{\pi }\right)$$

Cophase represents the time lag between joint movements on the modal frequency. In the case in which the cophase is close to zero degrees, it means that both joint movements are oscillating in the same direction (in-phase). If the cophase is close to 180 degrees, it indicates that both joint movements are oscillating in the opposite direction (anti-phase). The spatial data were cleaned using a low-pass filter with a cutoff frequency of 5 Hz and then differentiated to obtain the velocity–time domain.

### 2.5. Statistical Analysis

We conducted normality tests for variables, including dribbling accuracy, consistency, and three combinations of arm coordination. As a result, all variables satisfied the parametric assumption of normal distribution. Then, we used two-way (2 groups (novice or expert) × 3 tempos (normal, medium, or fast)) repeated measures ANOVA for dribbling accuracy and consistency. For arm coordination, the Harrison–Kanji test, known as the two-way repeated measures ANOVA, was used, which was provided from the circular statistics [20], because the cophase values were angular data. Post-hoc analyses were conducted using Watson–Williams tests, known as *t*-tests. All significance levels were set based on Bonferroni correction.

## 3. Results

### 3.1. Demographic Results

Table 1 shows the demographic results. Both groups had a similar age range, height, weight, and right arm length (*p* > 0.05).

### 3.2. Dribble Accuracy and Consistency

For dribble accuracy, no significant effects of group, tempo, or interaction were found (Figure 3A,C). For dribble consistency, significant main effects of group and tempo were found. Figure 3B represents group differences in the ML and AP directions (ML: *F*_(1, 15)_ = 8.28, *p* < 0.05, *η*^2^ = 0.389; AP: *F*_(1, 15)_ = 25.45, *p* < 0.001, *η*^2^ = 0.662). Darker gray bars stand for the expert group, while white bars stand for the novice group. In both directions, the expert group showed better dribble consistency compared to the novice group. Figure 3D represents the main effects of tempo in the ML and AP directions (AP: *F*_(2, 30)_ = 10.99, *p* < 0.001, *η*^2^ = 0.458).

Post-hoc analysis revealed that the consistencies with medium (*p* < 0.05) and fast tempos (*p* < 0.01) were significantly less relative to normal tempo (Figure 3D). Figure 3E,F show that there was no interaction effect between group and tempo on the accuracy and consistency of dribbling.

### 3.3. Upper Limb Coordination Patterns

The coordination pattern was determined by cophase values, which represent integrally the dribbling motion of the upper limb in the time–space domain. In general, a zero degree cophase value means that the signals are coupled in-phase, whereas the anti-phase mode would be reflected in 180 degrees.

Figure 4 represents three combinations of cophase: elbow–wrist (EW), wrist–finger (WF), and elbow–finger (EF) coordination. All cophase results in all plots showed negative values. The cophase values were calculated by subtracting the angle of a proximal joint from the angle of a distal joint. Thus, if the cophase value is positive, distal joint movement leads to proximal joint movement, while if the cophase value is negative, proximal joint movement leads to distal joint movement. Therefore, it can be seen that the result of a negative cophase indicates that the dribble motor skill shows a coordinated pattern in which the proximal joint leads.

Figure 4A shows the results of the main effects of group among three combinations of cophase. In the EW coordination, we found a significant main effect of group (*F*_(1, 39)_ = 5.76, *p* < 0.05, *η*^2^ = 0.129), with being closer to in-phase in the expert group compared to the novice group. Significant main effects of group also were found in WF (*F*_(1, 39)_ = 12.03, *p* < 0.001, *η*^2^ = 0.345) and EF (*F*_(1, 39)_ = 33.88, *p* < 0.001, *η*^2^ = 0.464) coordination. However, the main effects of group and interaction effect (group × tempo) were not significant (Figure 4B–D). These findings suggest that, regardless of the level of dribbling speed, the cophase in the expert group was closer to in-phase coordination, while the cophase in the novice group was closer to anti-phase coordination.

## 4. Discussion

The main purpose of this study was to compare dribbling performance according to proficiency and tempo and to apply the results to the field of motor learning. We examined two key factors: performance accuracy and motor coordination. The interpretation of the results is as follows.

Dribble accuracy did not differ between groups and dribble speeds, but there was a significant difference in performance variability. Regarding variability, the novice group exhibited higher levels compared to the skilled group when performing tasks accurately. The novice group also faced difficulties in task execution, with increased variability based on tempo, indicating challenges in the process of dribbling. Typically, it is known that higher levels of performance accuracy or motor expertise are associated with lower levels of variability in closed motor skills [21]. Novices can be characterized by high variability in motor performance due to the absence of a well-established motor pattern suitable for task execution and the occurrence of various unconsolidated movements depending on the situation, which is in line with previous studies that have reported high variability in novice performers [22].

According to the interpretation of the dual-control theory, the lack of significant differences in performance accuracy between the different levels of skill may be attributed to the issue of dual-control. Generally, it is known that dual-task performance is inferior to single-task performance due to the limited capacity of attentional resources [23]. Performing the task required participants to maintain precision on a small target while simultaneously synchronizing with a metronome’s specific tempo, which may have caused difficulties in dribble control. Difficulties in performing the task and limitations in attention capacity may lead to a cognitive burden for experts when presented with multiple instructions [24,25]. Therefore, both novice and skilled performers faced cognitive demands in this task, highlighting that the stability of dribbling performance is affected by factors such as skill level and tempo rather than just the accuracy of the bounce point.

One interesting observation is that cognitive load had a greater impact on consistency than on accuracy. The study found that novice dribbling consistency was lower compared to that of experts, which suggests that at higher levels of proficiency, it is more beneficial to control consistency than accuracy. In terms of dribbling consistency, the novice group demonstrated poor performance in both the forward and backward directions along the anteroposterior axis in accordance with tempo, suggesting that maintaining dribbling consistency in the AP direction becomes more difficult as task complexity increases. Thus, to improve the dribbling consistency of beginners in motor learning, it is crucial to provide feedback that emphasizes control of the AP direction. Although our study differed from previous research in terms of accuracy, which found that skill level and task difficulty (tempo) influence motor accuracy [2,26], it is crucial to consider overall performance excellence. To evaluate motor performance properly, it is necessary to analyze movement stability during task execution, as well as performance outcome [27].

Dribble accuracy was achieved through the movement of the right arm. Through an examination of kinematics in human movements, it may be possible to gain a more comprehensive understanding of how motor coordination affects performance accuracy. In this study, the cophase values indicate the time delay between joint movements at the modal frequency. According to the findings of the study’s cophase analysis, all values obtained during the task were negative, suggesting that the distal joint was leading the proximal joint on the phase plane. In general, the proximal-to-distal (PTD) sequence refers to the sequence of movements in joints or segments of the body, starting at the proximal (closer to the center of the body) and ending at the distal (farther from the center of the body) [28,29]. However, another study found that for motor skills requiring precise control, such as basketball dribbling, the distal joint (e.g., the hand) takes priority over the proximal body segment (e.g., the arm), which is different from the traditional PTD sequence observed in actions, such as the swing motion, that require maximum speed.

The tempo of the movement, which was the topic of interest in this study, was viewed as a control parameter [30] that modified the coordination pattern by altering the speed of the movement. We adjusted the dribble speed to examine how the pattern differed according to proficiency. The results showed that the expert group demonstrated an in-phase coordination pattern with cophase values close to zero, while the novice group exhibited an anti-phase pattern. These results are consistent with the study by Broderick and Newell [14], which showed differences in coordination patterns according to proficiency. However, the absence of interaction effects according to tempo suggests that the coordination pattern in the dribbling task was a stable characteristic that was not affected by movement speed. Previous studies on coordination patterns have focused on observing rapid changes in human intrinsic dynamics by asking participants to perform movements as quickly as possible [31,32,33]. However, the dribbling task in this study is a complex task that involves multiple joints in the body and is not commonly performed at maximum speed in real-life situations. Therefore, no changes in coordination patterns, such as phase transition, were observed within the 80% speed range of the maximum speed used in this task, and this was found to not be affected by proficiency.

The difference in the coordination pattern according to proficiency can be explained by the fact that when the movement skill level is low, there is an independent coordination structure in which joint correlations are not synchronized, resulting in an anti-phase tendency [13]. However, experts may have used a strategy of fixing the degree of freedom in the dynamics and reducing variability in arm movement. This is different from the strategy of fixing degrees of freedom and reducing variability in less-skilled performers, as mentioned by Bernstein [34]. In the case of repeated simple tasks, it is more important to use a strategy of fixing degrees of freedom to increase consistency, which may explain the results observed in this study.

The limitations of this study include the limited number of tempo conditions (three), which makes it difficult to reflect a wide range of situations in numerous spectra. Additionally, actual basketball dribbling involves considering numerous reactions to defense, but this study focused only on simple static dribbling. Therefore, to apply this to real sports situations, various settings and discussions regarding different situations are necessary. Furthermore, this study examined dribbling accuracy, variability, and motor coordination. However, future research should investigate a broader range of physiological characteristics, such as energy efficiency of movements for assessing proficiency, as well as psychological characteristics such as anxiety and arousal.

This study has academic significance in that it goes beyond the traditional approach of evaluating motor proficiency based solely on the results of performance and explores the qualitative characteristics of performance in conjunction with its outcomes. Furthermore, the attempt to elicit internal dynamic changes in humans through tempo control provides a scientific foundation that can contribute to the development of motor performance by exploring the principles of important control features in the domain of motor control. From a practical perspective, the results of this study provide evidence for the dimension (direction) of movements that coaches or learners in sports settings should focus on when practicing movements that require precise control, such as basketball dribbling.

## 5. Conclusions

Basketball dribbling should be performed accurately and consistently in various directions. Through this study, it was found that although there was no difference in dribbling accuracy between skilled and novice players, skilled players exhibited higher consistency in the AP direction, which was related to the in-phase coordination pattern. These results indicate that the study analyzed not only the outcomes but also the kinematic characteristics of the motor performance process. However, further analysis of additional variables in the psychological and physiological domains seems necessary for future research, and there appears to be a need to diversify the task conditions. Furthermore, additional efforts are required to apply the findings of this study to various sports disciplines, populations, and real-life sports settings.

## Figures and Tables

**Figure 1 ijerph-20-05788-f001:**
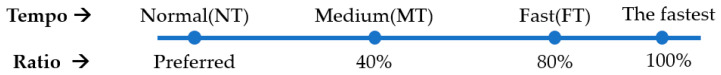
Tempo setup depending on individual dribble speed.

**Figure 2 ijerph-20-05788-f002:**
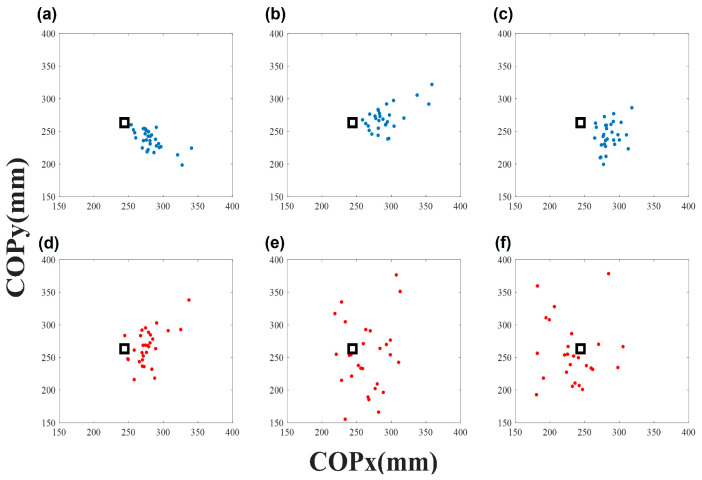
Distribution of dribble displacement on the force plate. The blue dots indicate the expert contact points for the target (black square), and the red dots indicate novice performance. The left column is the normal tempo (**a**,**d**), the middle column is the medium tempo (**b**,**e**), and the right column is the fast tempo (**c**,**f**). In the bouncing red point, it can be seen that it is more scattered than the blue dots.

**Figure 3 ijerph-20-05788-f003:**
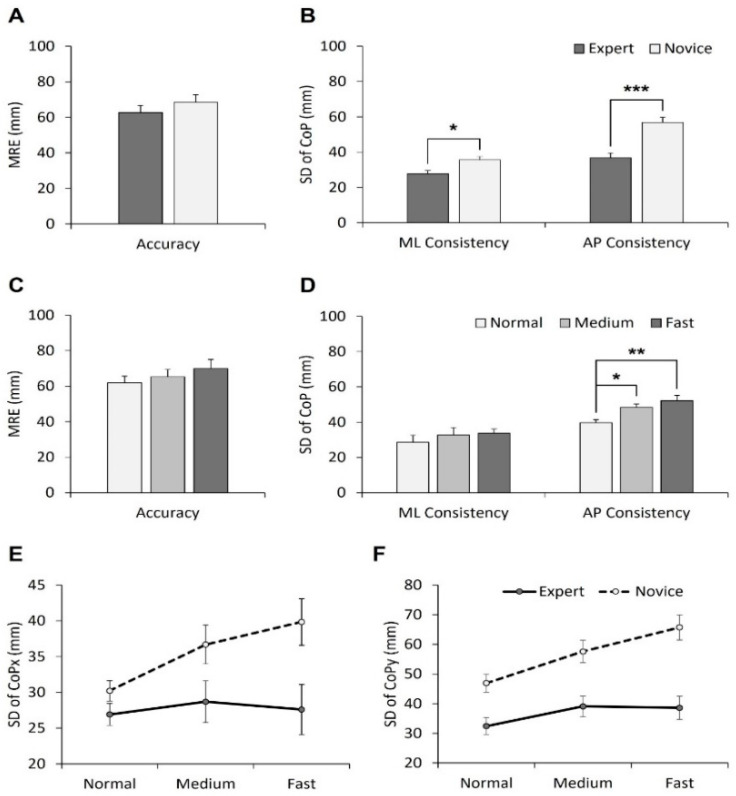
The accuracy and consistency of dribbling according to group (**A**,**B**), tempo (**C**,**D**), and interaction (**E**,**F**). (* MRE: mean radial error, COP: center of pressure, ML: medial–lateral, AP: anterior–posterior). * *p* < 0.05, ** *p* < 0.05, *** *p* < 0.001.

**Figure 4 ijerph-20-05788-f004:**
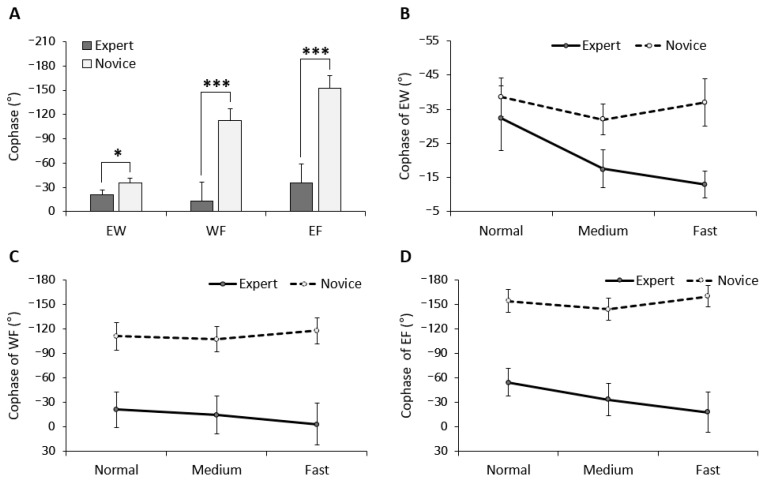
Results of cophase according to three combinations of arm coordination. Group differences in cophase (**A**) and tendency of cophase based on tempo (**B**–**D**). (* EW: elbow–wrist, WF: wrist–finger, EF: elbow–finger). * *p* < 0.05; *** *p* < 0.001.

**Table 1 ijerph-20-05788-t001:** Demographic results of expert and novice groups.

	Expert	Novice	*t*	*p*	Hedges’ g
Sample (*n*)	8	8			
Age (years)	23.12 (2.95)	26.57 (3.26)	−2.15	0.05	−1.05
Height (cm)	180.62 (5.53)	175.57 (3.69)	2.05	0.06	1.00
Weight (kg)	78.43 (6.33)	75.21 (4.12)	1.84	0.15	0.95
Right arm length (cm)	75.19 (5.15)	73.14 (3.44)	0.89	0.39	0.43

## Data Availability

Not applicable.

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
