# Peer review of "Dribble Accuracy and Arm Coordination Pattern According to Motor Expertise and Tempo"

_ijerph, 2023, doi:10.3390/ijerph20105788_

Round 1
Reviewer 1 Report
First of all, I would like to congratulate the authors for this great work. There are some aspects that need to be remedied before making a decision:
Abstract
I would include, before the objectives, some background information (the abstract is often presented alone and it would be good to include this information).
Introduction
In the introduction it is stated that theoretical models of motor control must be generated to study DOF and that this has not been done to analyze dribbling. Is there any reference model for other actions in other sports that can clarify this information and provide more data?
The present work includes hypotheses, but where are the aims of the work?
Methods
It would be appropriate to put the design of the study, the type of sampling used, the calculation of the sample size. In addition, other aspects should be included, such as the years of experience of the participants, or the differences at the sporting level, if any.
In addition, it has been indicated that informed consent was signed, but was the study approved by any ethics committee? Did it follow the Declaration of Helsinki?
The hight static dribble has been used in previous research? If so, reference them.
What is meant by sufficient warm-up exercises, please explain.
There was some kind of control of variables such as: time of measurement, last training performed, time since last training, etc. These are contaminating variables to take into account because they could affect the results.
The radial error is defined in the abstract, but not in the method, which is where it corresponds (data analysis).
Results
It would be appropriate that the information that the white bars belong to novices and the black bars to experts be included in the graphs themselves (there are times when they are presented alone without the article).
Discussion
You make two assertions from line 267 to 282, but do not justify what they may be due to. That is, you do not give an explanation why the consistency of novice dribbling was lower...
Lines 299 to 301: Previous studies on coordination patterns have focused on observing rapid changes in human intrinsic dynamics by asking participants to perform movements as quickly as possible. Where is the reference of these previous studies? What are these articles?
It seems to me that you have done a good job but I am missing more previous theoretical justification in the discussion, as well as a section with the limitations of the research and the possible practical applications of the results obtained.
Author Response
Thank you very much for providing such important and critical comments on my paper. I truly appreciate your insights and suggestions, and I have made every effort to respond to them and revise my paper accordingly. Attached, please find the revised version of my paper along with a summary of the modifications made. Thank you again for your valuable feedback and assistance.
Abstract
I would include, before the objectives, some background information (the abstract is often presented alone and it would be good to include this information).
I have added the additional background information of motor learning on lines 10 to 12.
Introduction
In the introduction it is stated that theoretical models of motor control must be generated to study DOF and that this has not been done to analyze dribbling. Is there any reference model for other actions in other sports that can clarify this information and provide more data?
I have added the relevant references in lines 61 to 64.
The present work includes hypotheses, but where are the aims of the work?
I have added the aim of this study in lines 90 to 93.
Methods
It would be appropriate to put the design of the study, the type of sampling used, the calculation of the sample size. In addition, other aspects should be included, such as the years of experience of the participants, or the differences at the sporting level, if any.
I have added the rationale for the number of study participants based on statistical analysis in lines 114 to 115 and, the average experience of basketball was in line 119.
In addition, it has been indicated that informed consent was signed, but was the study approved by any ethics committee? Did it follow the Declaration of Helsinki?
The study was conducted in accordance with the Declaration of Helsinki, and approved by the Institutional Review Board of Seoul National University (SNU IRB No. 1707/003-008).
The high static dribble has been used in previous research? If so, reference them.
The study on motor control was conducted using a static dribble task in the following paper: “Robalo, R. A. M. (2020). Informational variables for basketball dribble control.”
What is meant by sufficient warm-up exercises, please explain.
We performed a 3-minute warm-up stretching on the wrists, ankles, knees, hips, waist, and shoulders.
The radial error is defined in the abstract, but not in the method, which is where it corresponds (data analysis).
The information about RE is explained in formula 1 in line 170
There was some kind of control of variables such as: time of measurement, last training performed, time since last training, etc. These are contaminating variables to take into account because they could affect the results.
All measurements were taken at the same time interval.
Results
It would be appropriate that the information that the white bars belong to novices and the black bars to experts be included in the graphs themselves (there are times when they are presented alone without the article).
Legends have been added to all Figures.
Discussion
You make two assertions from line 267 to 282, but do not justify what they may be due to. That is, you do not give an explanation why the consistency of novice dribbling was lower...
Explained in lines 25 to 264
Lines 299 to 301: Previous studies on coordination patterns have focused on observing rapid changes in human intrinsic dynamics by asking participants to perform movements as quickly as possible. Where is the reference of these previous studies? What are these articles?
I added Reference [31-33], which explains the changes in coordination as changes in internal dynamic coordination patterns
It seems to me that you have done a good job but I am missing more previous theoretical justification in the discussion, as well as a section with the limitations of the research and the possible practical applications of the results obtained.
I have added information on dual-control theory and proximal-to-distal (PTD) sequence to line 265. Limitations added in lines 327-335 of the discussion.

Reviewer 2 Report
Dear Authors,
It's an interesting manuscript. I have some comments/questions.
It is worth considering weight and body composition in this study. Only height is given, but these variables significantly impact fine body management, especially in tall individuals.
Was it checked whether the subjects consumed auxiliary substances (e.g. caffeine) before the measurements? Has the nutrition and supplementation of these people been evaluated?
It is worth adding the limitations of the study.
Was the accuracy affected by the short duration of the exercise? What made you decide on this study protocol?
Line 259-266 - The results are surprising, the more so that different results are available in the literature. I think it's worth reflecting on what led to this situation. What could have influenced such a result and discuss it?
More details describe the experience of the players and the second group. How many years have they been training, what sports level do they represent, and what experience do they have? This is also crucial. And just like above, their body weight should be indicated, and nutrition and supplementation should be assessed.
It should be indicated what these results bring to science. I don't see a gap to be filled. It is necessary to supplement the knowledge about the participants' experience and characteristics, indicating the training period in which the research was carried out. This will help you repeat the test. It would be interesting to calculate the sample size. With such a small number of people, this is often done. Figure no. 2 is hardly legible.
Figure 3 - add explanations of abbreviations.
Statistically, significant differences should be discussed, which are indicated, for example, in figure 3. The discussion, although practical, does not strictly refer to the results. It should be indicated, especially those concerning significantly static changes.
In the abstract, data from statistics (statistically significant differences) should be added.
Conclusions should include information on limitations and directions for future research. These conclusions are extremely optimistic and a bit over the top.
We should discuss the results and draw more critical conclusions. The current results cannot be transferred to a larger group and applied.
Kind regards,
Reviewer
Author Response
Thank you very much for providing such important and critical comments on my paper. I truly appreciate your insights and suggestions, and I have made every effort to respond to them and revise my paper accordingly. Attached, please find the revised version of my paper along with a summary of the modifications made. Thank you again for your valuable feedback and assistance.
--------------------------------------------------------------------------
Only height is given, but these variables significantly impact fine body management, especially in tall individuals.
added the relevant information to Table 1.
Was it checked whether the subjects consumed auxiliary substances (e.g. caffeine) before the measurements? Has the nutrition and supplementation of these people been evaluated?
Participants were instructed to restrict their intake of caffeinated foods and beverages starting from the day before, and were restricted to those who did not take any special supplements.
It is worth adding the limitations of the study.
I have added in lines 327-335 of the discussion.
Was the accuracy affected by the short duration of the exercise? What made you decide on this study protocol?
This study examined the characteristics of control over a short period, as opposed to measuring changes in accuracy through specific exercises for learning, in order to confirm differences in motor performance. Relevant references are as follows.
Soderstrom NC, Bjork RA. Learning versus performance: an integrative review. Perspect Psychol Sci. 2015;10(2), 176–199. pmid:25910388
Statton, M. A., Encarnacion, M., Celnik, P., & Bastian, A. J. (2015). A Single Bout of Moderate Aerobic Exercise Improves Motor Skill Acquisition. PloS one, 10(10), e0141393.
McDonnell MN, Buckley JD, Opie GM, Ridding MC, Semmler JG. A single bout of aerobic exercise promotes motor cortical neuroplasticity. J Appl Physiol. 2013;114(9), 1174–1182. pmid:23493367
Line 259-266 - The results are surprising, the more so that different results are available in the literature. I think it's worth reflecting on what led to this situation. What could have influenced such a result and discuss it?
I have added in lines 254-275 of the discussion.
More details describe the experience of the players and the second group. How many years have they been training, what sports level do they represent, and what experience do they have? This is also crucial. And just like above, their body weight should be indicated, and nutrition and supplementation should be assessed.
I have added the average experience of basketball in rows 119.
It should be indicated what these results bring to science. I don't see a gap to be filled. It is necessary to supplement the knowledge about the participants' experience and characteristics, indicating the training period in which the research was carried out. This will help you repeat the test. It would be interesting to calculate the sample size. With such a small number of people, this is often done. Figure no. 2 is hardly legible.
The meaning of these results has been incorporated into the discussion, and the characteristics of the participants have been supplemented in line 119. Additionally, statistical justification for the sample size is presented in lines 114-115. Furthermore, an explanation of the scatterplot in Figure 2 has been added to aid reader comprehension.
Figure 3 - add explanations of abbreviations.
It has been supplemented
In the abstract, data from statistics (statistically significant differences) should be added.
I have added it to the abstract.
Conclusions should include information on limitations and directions for future research. These conclusions are extremely optimistic and a bit over the top.
I made revisions and additions to the limitations and practical implications in the conclusion, as well as clarified the significance of the findings reported in the paper.

Round 2
Reviewer 1 Report
The authors have resolved all the suggestions made.